# Sleep Quality and Patient Activation in Chronic Disease: A Cross-Sectional Mediation Analysis

**DOI:** 10.3390/clockssleep7030044

**Published:** 2025-08-22

**Authors:** Christian J. Wiedermann, Verena Barbieri, Stefano Lombardo, Timon Gärtner, Klaus Eisendle, Giuliano Piccoliori, Adolf Engl, Dietmar Ausserhofer

**Affiliations:** 1Institute of General Practice and Public Health, Claudiana—College of Health Professions, 39100 Bolzano, Italy; 2Provincial Institute for Statistics of the Autonomous Province of Bolzano—South Tyrol (ASTAT), 39100 Bolzano, Italy; 3Directorate, Claudiana—College of Health Professions, 39100 Bolzano, Italy; 4Claudiana Research, Claudiana—College of Health Professions, 39100 Bolzano, Italy

**Keywords:** patient activation, sleep quality, chronic disease, mediation analysis, self-management

## Abstract

Patient activation enhances self-management of chronic illnesses, and sleep quality is vital for health. The link between activation and sleep quality and the mediating role of chronic diseases remain underexplored. This study examined the association between patient activation and sleep quality, variations across chronic disease groups, and whether chronic diseases mediate this relationship. A population-based cross-sectional survey in South Tyrol (Italy) included 2090 adults (55.0% response rate). Patient activation was measured using the Patient Activation Measure (PAM-10), and sleep quality was measured using the Brief Pittsburgh Sleep Quality Index (B-PSQI). The presence and number of chronic diseases were self-reported. Bivariate analyses, multiple linear regression, and mediation analyses (PROCESS) were performed. Among the participants, 918 (44%) reported at least one chronic disease. These individuals had poorer sleep (B-PSQI mean: 5.05 ± 3.26 vs. 3.66 ± 2.65; *p* < 0.001) and lower patient activation (PAM-10: 54.4 ± 12.7 vs. 57.2 ± 12.5; *p* < 0.001) than those without. A negative correlation between PAM-10 and B-PSQI was observed (r = −0.12, *p* < 0.001), with stronger associations in patients with hypertension and mental illness. In adjusted regressions, chronic disease, female sex, and older age predicted poorer sleep, whereas higher PAM-10 scores predicted better sleep. Mediation analyses showed that chronic disease partially mediated the relationship between patient activation and sleep quality, accounting for 4.7% to 6.3% of the total effect. Conclusions: Higher patient activation correlates with better sleep quality, although this relationship is partly mediated by the chronic disease burden. Sleep disturbances persist across chronic conditions, despite good self-management. These findings highlight the importance of adopting strategies to manage chronic diseases and sleep disturbances, acknowledging that while patient activation is statistically associated with sleep quality, the strength of this relationship is limited.

## 1. Introduction

Patient activation involves an individual’s knowledge, skills, and confidence in managing their health and actively engaging in their healthcare management. Higher patient activation is linked to better self-management behaviors (e.g., healthy diet, exercise, and medication adherence) and improved clinical outcomes in chronic conditions [1,2]. More activated patients often have better diabetes control (lower HbA1c, improved blood pressure, and lipid levels) and other health indices [3].

Sleep quality is vital for health-related quality of life. Poor sleep impairs physical and mental well-being [4,5,6,7] and is linked to elevated blood pressure, slower recovery from illness and increased cardiovascular risk [8,9]. Thus, higher patient activation may correlate with better sleep quality, as activated patients may maintain healthier routines and effectively manage sleep-disrupting symptoms.

Several studies have assessed both patient activation and sleep outcomes; however, their findings are limited and mixed. A large cohort study of childhood cancer survivors (a population with significant chronic health burdens) found no significant association between survivors’ activation levels and sleep quality; the prevalence of clinically relevant sleep disturbances was similar in those with low activation (9.3%) versus high activation (9.8%), and overall sleep disturbance did not differ across activation level quartiles (as measured by the Health Activation Measure—PAM) [10]. Similarly, in a study of hospitalized patients with heart failure, higher activation was associated with better self-care and perceived control but not with better sleep; patient-reported sleep disturbance scores did not significantly differ between the low- and high-activation groups [9]. It should also be noted that subjective sleep quality, as measured by the PSQI, does not always linearly reflect the clinical disease burden. For example, even stroke patients in the subacute or chronic phase, despite having multiple chronic conditions, can report satisfactory sleep quality scores on the PSQI, highlighting the complex, multifactorial nature of perceived sleep and the limitations of patient-reported outcome measures (PROMs) in isolation [11]. These findings suggest that being an “activated” patient does not automatically translate to improved sleep quality.

Evidence suggests a possible connection between the two. A 2022 survey of adults with chronic conditions during the COVID-19 pandemic found that those with poor sleep were more likely to have low health activation levels (as measured by the Consumer Health Activation Index); 59% of patients with poor sleep had low activation, compared to approximately 50% among those with good sleep, a marginal but suggestive difference (*p* ≈ 0.05) [12]. Additionally, indirect evidence suggests that empowering patients may influence sleep; in hospital settings, patients who feel greater control over their care (related to activation) report longer sleep duration and better sleep quality [9].

Chronic health conditions can significantly affect a patient’s activation level [2,13] and their sleep quality. Many chronic diseases disrupt sleep through symptoms such as pain, nocturia, and breathing difficulties [4,12]. Managing a chronic illness often demands high patient engagement [1], making it essential to consider specific chronic conditions as potential mediators or moderators of the activation–sleep relationship. If patient activation affects disease management or severity and severity impacts sleep, it supports a mediation model in which chronic disease links activation to sleep quality. Understanding this is crucial for creating targeted interventions to improve health engagement and sleep outcomes in patients with chronic illnesses.

In recent years, patient activation has received increasing attention as a behavioral determinant of population health, particularly within integrated care frameworks and primary care transformation initiatives [14]. Understanding how activation interacts with sleep and chronic illness is particularly relevant, given the increasing emphasis on self-management and personalized care planning in chronic disease models.

Although highly activated patients generally achieve better health outcomes [2], a direct positive association between activation and sleep quality has not been established. Although prior studies have examined patient activation and sleep in specific clinical populations, research exploring this relationship across a general population with diverse chronic diseases remains scarce. In particular, the potential mediating role of chronic diseases in the activation–sleep link has not been systematically investigated.

This study aimed to address the following research question:Is higher patient activation associated with better sleep qualityWhich chronic diseases are most strongly associated with both patient activation and sleep quality?Does the presence of chronic disease mediate the relationship between patient activation and sleep quality?

This study contributes population-based evidence from a culturally distinct, multilingual European region, offering new insights into how activation and subjective sleep quality interact in the context of chronic diseases.

## 2. Results

### 2.1. Sample Characteristics

The characteristics of the study population are presented in Table 1. The sex distribution was balanced, with no statistically significant difference between individuals with and without chronic disease. In contrast, the age distribution differed significantly: individuals with chronic disease were more often in older age categories, while younger age groups were more represented among those without chronic disease.

A chi-square test revealed a significant overall association between the language background and chronic disease status (*p* < 0.05). Post hoc comparisons (adjusted standardized residuals) indicated that individuals with Italian as their first language were overrepresented among participants with chronic diseases, whereas German speakers were more frequently found in the group without chronic disease. A similar pattern was observed for citizenship: participants with non-Italian citizenship were disproportionately represented in the group without chronic diseases. No significant association was found between the municipality of residence (rural vs. urban) and chronic disease status.

Living arrangements differed according to chronic disease status. Living alone was significantly more common among those with chronic diseases, whereas those living with children or parents were more likely to be without chronic conditions. Educational attainment showed a strong inverse association with chronic disease status: individuals with only middle school education were overrepresented in the chronic disease group, while university graduates were more prevalent among those without a chronic disease.

Health status scores, categorized as “poor or moderate” versus “good or very good”, demonstrated a strong association with chronic disease. Most individuals with chronic conditions reported poor or moderate health, whereas those without chronic conditions predominantly reported good or very good health.

The sleep quality and patient activation scores differed significantly between the groups. The chronic disease group exhibited poorer sleep quality, as reflected by a higher B-PSQI score, whereas individuals without chronic disease had lower scores, indicating better sleep quality (mean ± SD, 5.0 ± 3.26 vs. 3.7 ± 2.65). With a Cliff’s delta of 0.197 (small effect size), the difference, although significant, was small in magnitude. Similarly, patient activation scores (PAM-10) differed significantly, with lower activation observed among individuals with chronic diseases (mean ± SD, 54.4 ± 12.71 vs. 57.2 ± 12.51). The effect size (Cliff’s Delta = −0.111) was negligible. However, these statistical significances are unlikely to be clinically meaningful.

### 2.2. Patient Activation and Sleep Quality Across Chronic Disease Groups

Sleep quality (B-PSQI) was generally poorer among individuals with chronic diseases than among those without, with the highest mean scores observed in patients with renal disease, liver disease, and mental illnesses. Patient activation (PAM-10) scores varied across disease groups, with individuals affected by immune system diseases reporting the highest levels of activation, whereas those with mental illnesses exhibited the lowest levels (Table 2).

A negative correlation between PAM-10 and B-PSQI scores was observed across most chronic disease groups, indicating that higher patient activation is generally associated with better subjective sleep quality. However, the strength of this association was consistently small, with Spearman’s rho values ranging from −0.045 to −0.391. Statistically significant but small inverse correlations were found in patients with arterial hypertension (ρ = −0.178) and mental illness (ρ = −0.220). In individuals with liver disease, a stronger correlation was observed (ρ = −0.391), suggesting a moderate effect; however, this did not reach statistical significance due to the small sample size. In contrast, no meaningful associations were identified in patients with pulmonary, renal, immune, oncological, or metabolic diseases, where the correlations were weaker and non-significant. Overall, these findings suggest that the relationship between patient activation and sleep quality is modest and may vary by disease type, with stronger effects in conditions characterized by behavioral or neuropsychological components.

Figure 1 shows the mean differences in PAM-10 and B-PSQI scores associated with the presence of specific chronic diseases. Mental illness showed the strongest association with lower PAM-10 scores and higher B-PSQI scores, indicating poorer sleep quality. Other chronic conditions, including metabolic, oncological, renal, and liver diseases, were also associated with reduced patient activation. Immune system diseases appeared to have minimal or slightly positive associations with the activation scores.

Renal, liver, and cardiovascular diseases were associated with higher B-PSQI scores, reflecting poorer sleep. In contrast, oncological diseases showed a noticeable association with lower patient activation, and their relationship with sleep quality was smaller than that of other chronic conditions.

These findings highlight the varied burden of chronic diseases on self-management and sleep, with mental health conditions showing the most pronounced dual associations. They emphasized the importance of tailored interventions that consider both patient activation and sleep disturbances in individuals with chronic illnesses.

### 2.3. Determinants of Patient Activation and Predictors of Sleep Quality

#### 2.3.1. Regression Analysis of PAM-10 Score

Multiple linear regression analysis was performed to examine the association between sociodemographic and clinical characteristics and patient activation (PAM-10 score) (Table 3). The model was statistically significant (F = 10.059, *p* < 0.001) and explained 2.9% of the variance in the PAM-10 scores (adjusted R^2^ = 0.029).

Higher PAM-10 scores were significantly associated with having a university education, being female, and living with a partner. In contrast, lower patient activation was predicted by older age, the presence of chronic disease, urban residence, and living with children.

Collinearity diagnostics were performed to ensure the independence of the predictors in the multiple linear regression model. Variance Inflation Factor (VIF) values for all predictors ranged from 1.01 to 1.28, and the corresponding tolerance values ranged from 0.78 to 0.99, indicating the absence of problematic multicollinearity. All VIF values were well below the commonly accepted threshold of 5, and the tolerance values exceeded the minimum acceptable value of 0.2. The Durbin–Watson statistic was 1.986, indicating no evidence of autocorrelation in the residuals, and supporting the assumption of independence.

#### 2.3.2. Regression Analysis of B-PSQI Score

A stepwise multiple linear regression analysis was conducted to examine the association between sociodemographic, health, and behavioral variables and subjective sleep quality, as measured by the PSQI-B score (Table 4). The model was statistically significant (F = 25.248, *p* < 0.001), with an adjusted R^2^ of 0.075, indicating that 7.5% of the variance in sleep quality was explained by predictors.

Poorer sleep quality was significantly associated with the presence of chronic disease, female sex, and older age. Higher patient activation (PAM-10) predicted slightly better sleep quality (standardized β = −0.022, *p* < 0.001), although the effect size was negligible. Language background also played a role: participants who identified as Italian or Ladin/Other reported significantly higher B-PSQI scores than German speakers. In contrast, university education was associated with better sleep quality, suggesting a protective effect of higher educational attainment.

Collinearity diagnostics indicated no concerning multicollinearity among the independent variables, with all VIF values ranging from 1.01 to 1.26 and tolerance values between 0.79 and 0.99, which were well within the acceptable thresholds. Furthermore, the Durbin–Watson statistic was 2.033, confirming the independence of the residuals.

#### 2.3.3. Regression Analysis of Number of Chronic Diseases

Multiple linear regression analysis was performed to examine the association between sociodemographic and clinical characteristics and the number of chronic diseases (Table 5). The model was statistically significant (F = 224.822, *p* < 0.001) and explained 17.6% of the variance in the number of chronic diseases (adjusted R^2^ = 0.176).

A higher number of chronic diseases was significantly associated with older age and a lower PAM-10 score. All other demographic variables were not significant in the model.

Collinearity diagnostics were assessed to ensure the independence of predictors in the multiple linear regression model. VIF values for all predictors were approximately 1.01, with corresponding tolerance values of 0.989, indicating the absence of problematic multicollinearity. All VIF values were well below the commonly accepted threshold of 5, and tolerance values exceeded the minimum acceptable value of 0.2. The Durbin–Watson statistic was 1.910, indicating no evidence of autocorrelation in the residuals and supporting the assumption of independence of the variables.

#### 2.3.4. Mediation Analysis

Three mediation models were tested to explore the role of patient activation in the relationship between chronic disease and sleep quality. The mediation framework used to examine the indirect effect of chronic disease on sleep quality through patient activation, while adjusting for age, is shown in Figure 2.

In Model 1 (chronic disease → PAM-10 → sleep quality), patient activation partially mediated the relationship between chronic disease status and sleep quality, accounting for 5.6% of the total effect. Model 2 (number of chronic diseases → PAM-10 → sleep quality) showed a similar pattern, with patient activation mediating 6.3% of the total effect. In Model 3, the analysis was adjusted for age only. Here, the indirect effect of chronic disease on sleep quality via patient activation remained present (4.7%), while age also showed a small but significant indirect effect via patient activation (6.4%).

These findings support a modest but consistent mediating role of patient activation in the pathway from chronic disease burden to subjective sleep quality (Table 6).

## 3. Discussion

This study investigated the relationships between patient activation (PAM-10), sleep quality (B-PSQI), and chronic disease status in a population-based sample. The findings confirmed that individuals with chronic diseases experience significantly poorer sleep quality than those without chronic conditions. In contrast, no statistically significant differences in patient activation scores were observed between the groups based on bivariate comparisons. However, regression analyses revealed that the presence of chronic disease was independently associated with slightly lower activation levels after adjusting for sociodemographic factors such as age, sex, education, and living situation. This suggests that the chronic disease burden can modestly constrain self-management capacity, although the effect is small and context-dependent.

The effect of patient activation on sleep quality was statistically significant but negligible in magnitude, highlighting its limited explanatory power. Although a weak negative correlation was observed between patient activation and sleep quality, this relationship varied by disease type. Stronger inverse associations were found in individuals with arterial hypertension and mental illness, whereas no meaningful association was detected for other chronic conditions, such as pulmonary, metabolic, or oncological diseases, indicating disease-specific patterns in the activation–sleep interplay.

Mediation analysis suggested that chronic disease burden may partially account for the association between patient activation and sleep quality. However, the estimated indirect effects, which accounted for 4.7% to 6.3% of the total effect depending on the model specification, were not statistically significant. This suggests that while chronic diseases may reduce self-management capacity, their role in linking activation to sleep quality remains limited and should be interpreted with caution. Supporting patient activation may contribute to mitigating sleep disturbances, especially in individuals with multiple chronic conditions or advancing age.

### 3.1. Comparison with Existing Literature

The observed association between female sex and poorer sleep quality is consistent with findings from large epidemiological studies showing that women are more likely to experience insomnia, lighter sleep, and lower sleep efficiency than men [15]. Biological, hormonal, and psychosocial factors, including caregiving responsibilities, mood disturbances, and sleep-related anxiety may contribute to these differences. Although activation was slightly higher among female participants in our sample, this did not offset their increased risk of poor sleep, suggesting that targeted sleep interventions may need to address sex-specific barriers to restorative sleep.

Prior studies have consistently shown that greater self-management is linked to improved overall health outcomes [1,2]. The relationship between patient activation and sleep quality has produced inconsistent findings in previous studies. In this study, we found that higher patient activation was associated with better sleep quality, and this relationship was partially mediated by the presence or burden of chronic diseases. While activation independently predicted lower B-PSQI scores, its positive impact on sleep was somewhat reduced among individuals with chronic conditions, highlighting that, while statistically detectable, the relationship between activation and sleep may be modest and influenced by multiple overlapping factors.

The significant negative correlation observed in our study between patient activation and sleep quality among individuals with hypertension suggests that greater engagement in self-management may be particularly beneficial in mitigating disease-related sleep disturbances in this population.

Mental health conditions further complicate this relationship. Depression and anxiety are well-documented contributors to poor sleep quality [16], and individuals affected by these conditions often demonstrate lower levels of patient activation due to diminished motivation and reduced self-efficacy for health management [10]. Our findings confirm that mental illness is among the chronic conditions most strongly associated with impaired sleep, underscoring that improving activation alone may not be sufficient unless mental health needs are addressed simultaneously.

Unlike many earlier investigations that overlooked the mediating role of chronic disease, our mediation analyses showed that patient activation influences sleep quality both directly and indirectly. Although chronic disease partially mediates this relationship, activation independently predicts better sleep quality. These findings highlight the importance of integrated care approaches that address both chronic disease management and sleep health, while also supporting efforts to enhance patient activation as a direct contributor to improved sleep.

### 3.2. Interpretation of the Mediation Effect

Mediation analysis revealed that the presence and burden of chronic diseases partially explained the association between patient activation and sleep quality. While individuals with chronic diseases showed similar activation scores in unadjusted comparisons, regression models indicated that chronic diseases were independently associated with slightly lower activation levels after controlling for relevant covariates. This may reflect the constraining effects of chronic illness on self-management capacity, despite increased exposure to healthcare and support services [1,2]. Our findings suggest that patient activation may offer modest but independent benefits for subjective sleep quality improvement.

The strongest associations between patient activation and sleep quality were observed in individuals with hypertension and mental illness, suggesting that greater activation may help mitigate sleep disturbances in these populations [10]. In contrast, no significant association was found in individuals with pulmonary, re15nal, immune, oncological, or metabolic diseases, indicating that sleep quality in these conditions may be driven by factors less responsive to self-management [12].

These findings highlight the need for tailored interventions that integrate patient activation strategies with sleep health management, particularly in individuals with hypertension [17] and mental health conditions [16], where disease-related sleep disturbances persist, and standard self-management approaches alone may not sufficiently improve sleep outcomes.

Given the cross-sectional nature of this study, the directionality of the observed associations cannot be determined with certainty. It is plausible that lower patient activation may contribute to the development or worsening of chronic conditions due to reduced engagement in preventive and self-care behaviors. Similarly, sleep disturbances may both result from and contribute to chronic illness progression, suggesting potential feedback loops between activation, sleep, and disease burden.

### 3.3. Disease-Specific Considerations

The relationship between patient activation and sleep quality appears to vary by chronic disease, suggesting that disease-specific mechanisms may shape the effectiveness of self-management in improving sleep quality. Stronger negative associations were observed in individuals with hypertension and mental illness, whereas no significant associations were observed for pulmonary, renal, immune, oncological, or metabolic conditions.

Hypertension demonstrated one of the strongest associations between patient activation and sleep quality, consistent with existing evidence linking cardiovascular health and sleep disturbance [8,17,18]. Although higher activation is known to support medication adherence and healthy lifestyle changes, sleep quality in individuals with hypertension often remains compromised due to autonomic dysfunction, nocturnal blood pressure variability, and a high prevalence of sleep apnea [9]. These findings underscore the importance of integrated care strategies that simultaneously target blood pressure and sleep health.

Mental health disorders, particularly depression and anxiety, showed the strongest associations with poor sleep quality, consistent with their well-established links to insomnia, hyperarousal, and disrupted circadian rhythms [16]. Patients affected by these conditions also tend to exhibit lower levels of activation, likely due to reduced motivation, energy, and self-efficacy, which collectively hinder effective self-management [10]. These findings suggest that patient activation strategies may be insufficient on their own and should be complemented by targeted mental health interventions to improve sleep outcomes in this population group.

In contrast, no significant association between activation and sleep quality was observed among individuals with diabetes, despite the high prevalence of sleep disturbances related to neuropathy, nocturia, and glycemic variability [3,16]. It is possible that more highly activated patients achieve better glucose control, partially offsetting the sleep-related burden of the disease. However, given that more than half of people with diabetes report poor sleep, the severity and complexity of the disease may moderate or obscure the influence of self-management behaviors on sleep quality.

The absence of a significant patient activation–sleep association in several chronic conditions, such as pulmonary, renal, immune, oncological, and metabolic diseases, suggests that physiological factors such as pain, systemic inflammation, or medication side effects may be the primary drivers of sleep disturbances in these populations, overriding the influence of self-management behaviors [5,7]. This underscores the importance of future research focusing on the role of symptom control in improving sleep outcomes, potentially offering greater benefits than activation strategies alone in these groups.

An additional consideration is the role of pharmacological therapy, which may influence the association between chronic diseases and sleep quality. Conditions such as hypertension and mood/anxiety disorders are often managed with medications known to affect sleep, either positively (e.g., symptom control) or negatively (e.g., side effects such as sedation and insomnia). The stronger activation–sleep associations observed in these groups may reflect, in part, better disease control through medication, which can reduce the symptom burden and facilitate better sleep. Conversely, in conditions such as cancer or chronic respiratory disease, pharmacological management may be less consistent or associated with greater symptom persistence or side effects, possibly explaining the weaker associations observed in these conditions. As medication use was not assessed in this study, it remains an important area for future research.

The variability in the association between patient activation and sleep quality across chronic disease groups may reflect the underlying differences in pathophysiology and symptom expression. For example, in arterial hypertension, patient activation may promote adherence to lifestyle and pharmacological interventions that reduce nocturnal blood pressure fluctuations [19], which are important contributors to poor sleep [20]. In mental health disorders, especially anxiety and depression, both activation and sleep are closely tied to emotional regulation and cognitive processing, which could explain the stronger associations observed [21]. In contrast, for conditions such as cancer or chronic pulmonary disease, persistent symptoms such as pain, breathlessness, or fatigue, which are often less modifiable through self-management alone, may blunt the impact of activation on sleep. This heterogeneity underscores the need for disease-specific frameworks for interpreting patient activation–sleep interactions.

Overall, the findings suggest that individuals with hypertension and mental illness may benefit most from interventions that integrate targeted self-management support with behavioral strategies aimed at improving sleep. In contrast, for conditions in which activation appears less relevant to sleep outcomes, a stronger focus on medical symptom management may be warranted. Recognizing these disease-specific differences in the activation–sleep relationship can inform the development of tailored and more effective sleep interventions for individuals with chronic conditions.

### 3.4. Clinical and Public Health Implications

These findings support the consideration of integrated care strategies, as chronic diseases mediate the activation–sleep relationship. Interventions should address both self-management and disease-specific sleep disturbances, rather than relying on activation alone.

The strongest activation–sleep associations in patients with hypertension and mental illness suggest that these patients may benefit the most from integrated interventions. In patients with hypertension, combining self-management education with screening for sleep apnea and nocturnal hypertension could improve outcomes. For mental health conditions, addressing insomnia, stress, and emotional well-being, along with activation strategies, is essential.

While activation improves self-care, its effect on sleep quality is limited in conditions where physiological factors dominate (e.g., renal disease, cancer, and metabolic disorders). Symptom management and treatment optimization should be prioritized.

Chronic disease guidelines should incorporate sleep health, particularly when self-management alone does not improve sleep quality. Providers should recognize sleep disorders as key to chronic disease care and integrate behavioral sleep interventions when needed [9,14].

It should be noted that sleep quality is influenced by a wide range of psychosocial and lifestyle factors beyond patient activation and disease burden. For example, recent evidence from an Italian cohort study suggests that diets rich in anthocyanin-containing fruits are associated with a lower risk of poor sleep quality and depressive symptoms [22], highlighting the potential role of nutritional interventions in improving sleep health.

### 3.5. Limitations and Future Research Directions

The B-PSQI assesses subjective sleep quality, which may not directly correspond to objective clinical measures of disease burden or sleep disturbances, potentially limiting the precision of the observed associations. This study investigated the association between patient activation, chronic disease, and sleep quality, while acknowledging several limitations.

First, the cross-sectional design precludes the establishment of causality, rendering it uncertain whether lower activation contributes to poor sleep or whether poor sleep diminishes activation by impairing motivation and self-care. Longitudinal studies are required to elucidate these relationships. Future longitudinal research is also needed to examine the directionality of the effects between patient activation, chronic disease development, and sleep disturbance. An important limitation of this study is the use of cross-sectional data to conduct mediation analysis. While mediation models aim to capture causal pathways over time, our design does not allow for the temporal ordering of variables. As highlighted by Maxwell and Cole [23] and Maxwell et al. [24], cross-sectional mediation models, whether assuming complete or partial mediation, can produce biased or misleading estimates. In particular, a significant indirect effect observed in a cross-sectional model may not correspond to a true longitudinal mediation effect, even when partial mediation is assumed. Therefore, the mediation findings should be interpreted as conditional associations rather than causal mechanisms. Longitudinal data are needed to confirm whether patient activation mediates the relationship between the chronic disease burden and sleep quality.

Second, reliance on self-reported measures (PAM-10 and B-PSQI) may have introduced recall bias. Future research should incorporate objective sleep assessments (e.g., actigraphy and polysomnography) to enhance accuracy.

Third, chronic diseases were analyzed as binary variables or the number of chronic diseases, without considering their severity. More detailed data could elucidate how the disease burden influences activation and sleep quality, particularly in conditions such as hypertension and mental illness, in which stronger associations have been observed.

Fourth, the generalizability of our findings may be limited because the study population was exclusively from South Tyrol, a culturally distinct and multilingual region of Northern Italy. Health behaviors, patient activation, and sleep quality may differ in other geographic, ethnic, or healthcare system contexts, which should be considered when interpreting the results.

Additionally, interventional studies are lacking in this field. While the findings suggest that higher activation does not directly improve sleep, future trials should evaluate whether activation-enhancing programs (e.g., self-management education and e-health tools) impact sleep when combined with behavioral sleep interventions. Intervention trials should incorporate validated sleep measures along with PAM-10 assessments to determine whether improvements in activation lead to better sleep quality. If activation-enhancing programs improve both self-care and sleep, this would support a causal link. Conversely, if poor sleep impedes activation, addressing sleep disorders (e.g., insomnia and sleep apnea) may be necessary to enhance self-management.

Chronic disease status was assessed through self-report and not verified by clinical records or administrative data (e.g., ICD codes or exemption status), which may have introduced misclassification or reporting bias. The survey did not include data on specific therapies, such as CPAP use for sleep-disordered breathing, which limits the ability to account for the treatment effects on sleep quality in patients with respiratory or other chronic conditions.

Finally, disease-specific analyses are required to clarify which chronic conditions benefit most from activation-based versus symptom-focused interventions. Understanding these differences could refine integrated care approaches to improve both self-management and sleep quality.

## 4. Methods

### 4.1. Study Design

This cross-sectional study analyzed patient activation and sleep quality in individuals with and without chronic disease. It was jointly conducted by the Provincial Institute of Statistics (ASTAT; Istituto Provinciale di Statistica—Landesinstitut für Statistik) and the Institute of General Practice and Public Health in the Autonomous Province of Bolzano, South Tyrol, between 1 March and 30 May 2024.

The survey also included additional items on health literacy (HLS-EU-Q16), health information-seeking behavior, and health-related behaviors (HBC), the results of which are reported separately [25]. This study specifically focused on the relationship between sleep quality, patient activation, and chronic disease status.

### 4.2. Setting and Sample

South Tyrol, the Autonomous Province of Bolzano, is part of the Trentino–Alto Adige region in Italy (total population: 535,000), with approximately 70% German-speaking, 25% Italian-speaking, and 5% other languages. The target population of the survey was approximately 400,000 individuals aged 18 years and older, residing in South Tyrol. Stratified probabilistic sampling was used in this study. The ASTAT randomly selected adults aged ≥ 18 years, stratified by age (18–34, 35–54, 55, and above), sex (male and female), citizenship (Italian or other), and residence (municipality), from the register of the current resident population in the province. To ensure an adequate level of precision, 4000 individuals were sampled, considering the distribution and variation between the strata. Participants were required to complete the questionnaire independently; therefore, individuals with substantial cognitive or language impairments were implicitly excluded from participation.

### 4.3. Participant Survey

The participant survey was designed collaboratively by the ASTAT and the Institute of General Practice and Public Health. The German and Italian versions translated from ASTAT were reviewed for language equivalence by a research group at the Institute for General Practice and Public Health, Germany.

#### 4.3.1. Sleep Quality

Most epidemiological research on sleep quality depends on self-reported tools, such as the Pittsburgh Sleep Quality Index (PSQI) [26]. The original PSQI is a widely recognized self-report questionnaire that evaluates sleep quality and disturbances over one month [16]. It has been extensively used in both clinical and research contexts, proving its reliability and validity across various populations and conditions [27,28]. The original PSQI includes 19 items divided into seven components. We utilized the brief version (B-PSQI) described and validated by Sancho-Domingo et al. [29], who demonstrated its internal validity through confirmatory factor analyses and its sensitivity and specificity in identifying poor sleepers, akin to the full PSQI version. The brief version comprised six items from the original to evaluate five dimensions: perceived sleep quality, sleep duration, sleep efficiency, sleep latency, and sleep disturbance. These six questions of the B-PSQI resulted in five components rated on a scale of 0–3, similar to the original, with higher scores indicating more significant sleep disturbances or poorer sleep quality. The component scores were summed to create a global B-PSQI score ranging from 0 to 15, with scores above 5 suggesting poor sleep quality [29]. In our study, we employed six items from the original Italian and German PSQI versions. Reliability testing showed acceptable internal consistency for both language versions (Cronbach’s α = 0.77 for the Italian version [30], Cronbach’s α = 0.74 for the German version [31]).

#### 4.3.2. Patient Activation

Patient activation was assessed using the 10-item Patient Activation Measure (PAM-10), which evaluates individuals’ knowledge, skills, and confidence in managing their health [32]. The PAM-10 is a shortened version of the original 13-item PAM (PAM-13), which retains the 10 original items while maintaining reliability and validity. As no specifically validated PAM-10 versions are available in Italian or German, the corresponding 10 items from the validated PAM-13 versions in these languages were used in this study. These validated translations have demonstrated good psychometric properties, including high internal consistency and construct validity [33,34]. In an international comparison of the psychometric properties of four European language versions, including German, the reported Cronbach’s alpha values ranged from 0.80 to 0.88 [35].

Each item was rated on a 5-point Likert scale (1) strongly disagree, (2) disagree, (3) neutral, (4) agree, and (5) strongly agree. Raw scores were transformed into a standardized activation score ranging from 0 (lowest activation) to 100 (highest activation) following the established PAM scoring methodology.

#### 4.3.3. Sociodemographic Characteristics, Health and Chronic Diseases

Participants’ sociodemographic characteristics included age (birth year), gender (male/female), native tongue (German/Italian/Ladin/more than one or Others), citizenship (Italy/other country), educational level (primary school/vocational/high school/university), community and region of origin (rural/urban), and living situation (alone/with spouse, family member, parents, or children).

Health-related factors included self-reported health status (scale from 1 to 100) and the diagnosis of chronic disease. Health status was categorized as “Poor or Moderate” (<80) and “Good or Very Good” (≥80) based on a median score of 80 in the study population. Chronic conditions were recorded using multiple response categories covering nine specific diagnoses (e.g., cardiovascular, metabolic, and oncological). These variables were coded as independent dichotomous indicators, meaning that each diagnosis was treated as a separate binary variable. All health-related variables, except for living alone, which was a single item, were collected in a way that allowed multiple simultaneous indications.

### 4.4. Data Collection

Letters were mailed from ASTAT to randomly sampled participants to inform them about the study and invite them to voluntarily participate by completing the survey alone or with the aid of a family member or friend. The survey was completed through online self-completion or telephone interviews with ASTAT collaborators. One month after the first letter, a second letter was sent to inform the participants about the study and invite them to participate in it. An online survey was created using LimeSurvey [36].

### 4.5. Statistical Analysis

Analyses were conducted on complete data. Participants with missing data on any of the key variables, PAM-10, B-PSQI, chronic disease status, or covariates, were excluded, resulting in a final analytic sample of 2090 participants. All statistical analyses were conducted using IBM SPSS Statistics (version 27.00) with the PROCESS macro (version 4.2) for the mediation analysis. *p*-values were two-sided with a significance threshold of <0.05.

#### 4.5.1. Descriptive Statistics, Group Comparisons, and Correlation Analyses

Descriptive statistics were calculated for all sociodemographic, clinical, and behavioral variables of the participants. Continuous variables (e.g., PAM-10 and B-PSQI) were reported as medians with interquartile ranges (IQR) or means with standard deviations (SD). Categorical variables are reported as frequencies and percentages.

The chi-square test was used for categorical variables to examine the differences between individuals with and without chronic diseases. For non-normally distributed continuous variables, such as PAM-10 and B-PSQI scores, non-parametric Mann–Whitney U tests were used. Effect sizes for group differences were calculated using a rank-biserial correlation.

Spearman’s rank correlation coefficients (ρ) were calculated to assess the association between patient activation (PAM-10) and sleep quality (B-PSQI) across different chronic disease groups. Effect sizes for correlations were interpreted using Cohen’s guidelines: small (r ≈ 0.10), medium (r ≈ 0.30), and large (r ≥ 0.50) correlations were considered meaningful [37].

#### 4.5.2. Regression and Mediation Analyses

Two separate stepwise multiple linear regression models were constructed to examine the predictors of (1) patient activation (PAM-10) and (2) sleep quality (B-PSQI). In each step, the F-probability for inclusion was 0.05 and for exclusion, 0.1. Covariates for the regression analyses were selected a priori based on theoretical relevance and prior evidence linking sociodemographic factors and chronic disease status to both patient activation and sleep quality. The independent variables included sociodemographic factors (age, gender, education, language), health status (presence of chronic disease), and living situation. In the B-PSQI model, the PAM-10 score was included as a continuous predictor.

Model diagnostics included tests for multicollinearity (Variance Inflation Factor [VIF] and tolerance), normality and independence of residuals (Durbin–Watson statistic), and homoscedasticity. Regression coefficients (β), 95% confidence intervals, and *p*-values were reported for each predictor variable. The model fit was evaluated using R^2^ and adjusted R^2^. Standardized beta coefficients (β) from regression models were interpreted using conventional guidelines: values around 0.10 were considered small, around 0.30 were considered moderate, and 0.50 or higher were considered large in magnitude [38].

To assess whether the presence of chronic disease mediates the relationship between patient activation (PAM-10) and sleep quality (B-PSQI), a regression-based mediation analysis was conducted [23,24,25] using the PROCESS macro for SPSS (Model 4) with 1000 bootstrap resample. The analysis followed a standard mediation framework by estimating the total, direct, and indirect effects (Figure 3).

The following ordinary least squares (OLS) regressions were estimated:Mediator path (a): Chronic disease (binary or count) regressed on patient activation or mediator path (a): Patient activation regressed on chronic disease (binary or count)Mediation model (b + c′): Sleep quality regressed on both patient activation and chronic disease status.

The indirect effect (a × b) quantifies the extent to which chronic diseases explain part of the relationship between activation and sleep quality. Statistical significance was determined using bootstrapped 95% confidence intervals; mediation was considered present when the confidence interval did not include zero. To account for potential confounding factors, all models were adjusted for age.

## 5. Conclusions

This study explored the interplay between patient activation, chronic disease burden, and sleep quality in a large population-based sample. Although mediation analyses suggested that chronic diseases might account for a small proportion of the association between patient activation and sleep quality, these indirect effects were not statistically significant. This highlights the need for longitudinal research to establish potential causal pathways.

The strongest activation–sleep associations were observed in individuals with hypertension and mental illness, suggesting that these patient groups may particularly benefit from integrated interventions that combine self-management support with behavioral strategies aimed at improving sleep. In contrast, for many chronic conditions, especially those in which physiological symptoms predominate, sleep disturbances appear largely unrelated to activation levels, underscoring the importance of symptom-focused approaches.

Given the cross-sectional design and reliance on subjective measures, these findings should be interpreted cautiously. Future longitudinal and interventional studies are needed to clarify the causal direction of the activation–sleep relationship and to assess whether enhancing patient activation can meaningfully improve sleep outcomes, particularly in specific clinical populations. However, our findings suggest that patient activation alone is unlikely to substantially mitigate sleep disturbances in the presence of chronic diseases, highlighting the importance of combining activation strategies with direct symptom management and disease-specific interventions.

## Figures and Tables

**Figure 1 clockssleep-07-00044-f001:**
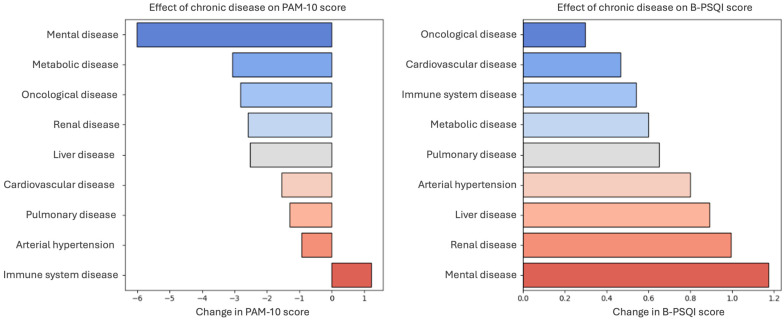
Impact of chronic disease on patient activation (PAM-10) and Sleep Quality (B-PSQI) scores. The figure illustrates the effects of various chronic diseases on patient activation (PAM-10) and sleep quality (BPSQI) scores. (**Left**) Change in the PAM-10 score, where negative values indicate a reduction in patient activation. (**Right**) Changes in BPSQI scores, with higher values reflecting poorer sleep quality.

**Figure 2 clockssleep-07-00044-f002:**
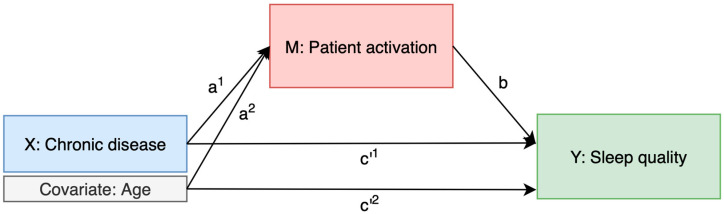
Mediation model of the relationship between chronic disease and sleep quality through patient activation, adjusted for age.

**Figure 3 clockssleep-07-00044-f003:**
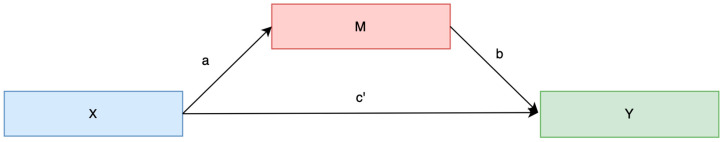
Mediation models of the relationships between patient activation, chronic disease, and sleep quality. The mediation model examined whether the presence of chronic disease or patient activation (M) mediates the relationship between patient activation (PAM-10) or chronic disease (X) and sleep quality (Y, B-PSQI). Path a: Effect of patient activation on chronic diseases. Path b: Effects of chronic disease or patient activation on sleep quality. Path c′ (Direct Effect): Effect of patient activation or chronic disease on sleep quality. Total Effect (c): The overall association between X and Y, which is composed of both the direct effect (c′) and the indirect effect (a × b).

**Table 1 clockssleep-07-00044-t001:** Sociodemographic, health, and behavioral characteristics of participants according to their chronic disease status.

Characteristic	Overall	No Chronic Disease	Chronic Disease	*p*-Value ^1^
Gender—% (*n*)				0.261
Female	1153 (55.2)	702 (60.9)	451 (39.1)	
Male	937 (44.8)	593 (63.2)	344 (36.7)	
Age (years)—% (*n*)				<0.001
18–34	383 (18.3)	324 (84.6)	59 (15.4)	
35–54	647 (31.0)	495 (76.5)	152 (23.5)	
55–99	1060 (50.7)	476 (44.9)	584 (55.1)	
Native tongue—% (*n*)				<0.001
German	1398 (66.9)	885 (63.3)	513 (36.7)	
Italian	499 (23.9)	270 (54.1)	229 (45.9)	
Ladin or Other	193 (9.2)	140 (72.5)	53 (27.5)	
Citizenship—% (*n*)				0.057
Italian	2011 (96.2)	1238 (61.6)	773 (38.4)	
Other	79 (3.8)	57 (72.2)	22 (27.8)	
Community—% (*n*)				0.079
Urban	381 (18.2)	221 (58.0)	160 (42.0)	
Rural	1709 (81.8)	1074 (62.8)	635 (37.2)	
Living situation—% (*n*)				
Alone	383 (18.3)	196 (51.2)	187 (48.8)	<0.001
With partner/family	1326 (63.4)	831 (62.7)	495 (37.3)	0.380
With children	785 (37.6)	556 (70.8)	229 (29.2)	<0.001
With parents	196 (9.4)	164 (83.7)	32 (16.3)	<0.001
With other family members	124 (5.9)	86 (69.4)	38 (30.6)	0.080
Educational level—% (*n*)				<0.001
Middle school or lower	492 (23.5)	211 (42.9)	281 (57.1)	
Vocational school (2–3 years)	674 (32.2)	430 (63.8)	244 (36.2)	
High School (5 years)	530 (25.4)	366 (69.1)	164 (30.9)	
University	394 (18.9)	288 (73.1)	106 (26.9)	
Health status score				<0.001
Poor or moderate	1245 (59.6)	614 (47.4)	631 (79.4)	
Good or very good	845 (40.4)	681 (52.6)	164 (20.6)	
B-PSQI Score—median (IQR) ^2^	3.0 (4.0)	3.0 (3.0)	4.0 (5.0)	<0.001 ^3^
PAM-10 Score—median (IQR) ^4^	52.9 (12.6)	52.9 (12.6)	52.9 (15.2)	<0.001 ^5^

^1^ Chi-square tests were used for categorical variables and Mann–Whitney U tests for continuous variables. ^2^ B-PSQI Score represents subjective sleep quality, with higher scores indicating worse sleep. ^3^ Effect size rank-biserial correlation: 0.197. ^4^ The PAM-10 Score represents patient activation, with higher scores indicating greater engagement in health management. ^5^ Effect size rank-biserial correlation: −0.111.

**Table 2 clockssleep-07-00044-t002:** Association between patient activation (PAM-10) and sleep quality (B-PSQI) across chronic disease groups.

Disease	*n*	B-PSQI ^1^ Mean (SD)	B-PSQI ^1^ Median (25–75%)	PAM-10 ^2^ Mean (SD)	PAM-10 ^2^ Median (25–75%)	Correlation Coefficient ^3^	*p*-Value
Pulmonary Disease	97	4.9 (3.52)	4.0 (2.0–7.0)	54.7 (14.83)	51.0 (47.4–62.6)	−0.087	0.339
Cardiovascular Disease	171	4.9 (3.41)	4.0 (2.0–7.0)	53.4 (12.18)	51.0 (45.1–59.3)	−0.172	0.025
Arterial Hypertension	399	5.0 (3.26)	4.0 (3.0–7.0)	54.4 (12.71)	52.9 (47.4–60.95)	−0.178	<0.001
Renal Disease	43	5.5 (3.40)	5.0 (2.5–7.5)	52.3 (15.90)	51.0 (45.1–62.6)	−0.191	0.220
Immune System Disease	73	4.8 (2.93)	4.0 (3.0–6.0)	57.6 (15.75)	59.3 (50.0–65.8)	−0.222	0.060
Oncological Disease	93	4.8 (3.32)	4.0 (2.0–7.0)	52.6 (10.92)	51.0 (47.4–59.3)	−0.045	0.668
Metabolic Disease	157	4.9 (3.34)	4.0 (2.0–7.0)	52.5 (12.77)	51.0 (45.1–59.3)	−0.078	0.334
Liver Diseases	24	5.2 (3.25)	4.0 (3.0–7.25)	53.2 (11.50)	50.0 (45.1–54.5)	−0.391	0.059
Mental Disease	122	5.4 (3.47)	5.0 (2.0–7.0)	50.6 (11.82)	50.5 (43.45–58.47)	−0.220	0.015
No Chronic Disease	1295	3.7 (2.65)	3.0 (2.0–5.0)	57.2 (12.51)	52.9 (50.0–62.6)	−0.113	<0.001

^1^ B-PSQI Score represents subjective sleep quality, with higher scores indicating worse sleep. ^2^ PAM-10 Score represents patient activation, with higher scores indicating greater engagement in health management. ^3^ Spearman’s correlation analysis for B-PSQI and PAM-10; Abbreviations: B-PSQI, Brief Pittsburg Sleep Quality Index; SD, standard deviation; PAM-10, Patient Activation Measure.

**Table 3 clockssleep-07-00044-t003:** Stepwise multiple linear regression analysis predicting the PAM-10 score.

Predictor	β	SE	t	95% CI		*p*-Value
				Lower	Upper	
(Constant)	58.131	1.047	55.513	56.078	60.185	<0.001
Age (continuous)	−0.052	0.017	−2.960	−0.086	−0.017	0.003
Education: University (vs. middle)	2.767	0.723	3.826	1.349	4.185	<0.001
Urban residence (Bolzano)	−2.222	0.722	−3.079	−3.637	−0.807	0.002
Chronic Disease (yes)	−1.692	0.617	−2.743	−2.902	−0.483	0.006
Gender (female)	1.535	0.557	2.758	0.444	2.627	0.006
Living with partner	1.564	0.597	2.621	0.394	2.735	0.009
Living with children	−1.278	0.599	−2.136	−2.452	−0.104	0.033

**Table 4 clockssleep-07-00044-t004:** Stepwise multiple linear regression analysis predicting B-PSQI scores.

Predictor	β	SE	t	95% CI		*p*-Value
				Lower	Upper	
(Constant)	3.73	0.362	10.299	3.018	4.437	<0.001
Chronic Disease	0.746	0.138	5.389	1.017	0.474	<0.001
Age (continuous)	0.016	0.004	4.212	0.009	0.024	<0.001
Gender (female)	0.628	0.125	5.040	0.384	0.873	<0.001
PAM-10 Score (continuous)	−0.022	0.005	−4.504	−0.032	−0.012	<0.001
Language: Italian (vs. German)	0.584	0.148	3.947	0.294	0.875	<0.001
Education: University (vs. middle)	−0.431	0.163	−2.641	−0.750	−0.111	0.008
Language: Other (vs. German)	0.665	0.279	2.381	0.117	1.213	0.017

**Table 5 clockssleep-07-00044-t005:** Stepwise multiple linear regression analysis predicting the number of chronic diseases.

Predictor	β	SE	t	95% CI		*p*-Value
				Lower	Upper	
(Constant)	−0.156	0.099	−1.577	−0.350	0.038	n.s.
Age (continuous)	0.020	0.001	20.157	0.018	0.022	<0.001
PAM-10 score	−0.006	0.001	−4.410	−0.009	−0.003	<0.001

**Table 6 clockssleep-07-00044-t006:** Mediation analysis results.

No.	Mediation Models	Total Effect	Direct Effect	Indirect Effect		
		(c = c′ + a × b)	(c′)	a	b	a × b (%)
1	X: Chronic disease (dichotomous) → M: PAM-10 → Y: Sleep quality	1.1204 ***	1.0581 ***	−2.5508 ***	−0.0244 ***	0.0623 (5.56%)
2	X: Chronic disease (number) → M: PAM-10 → Y: Sleep quality	0.6856 ***	0.6423 ***	−1.9166 ***	−0.0226 ***	0.0434 (6.32%)
3	X: Chronic disease (dichotomous) → M: PAM-10 → Y: Sleep quality (adjusted for Age)	0.8305 ***	0.7915 ***	−1.7257 **	−0.0226 ***	0.0391 (4.71%)
	Age → PAM-10 → Sleep quality	0.0202 ***	0.0189 ***	−0.0575 ***	−0.0226 ***	0.0013 (6.44%)

Path a represents the effect of the independent variable (X) on the mediator (M); path b represents the effect of the mediator (M) on the dependent variable (Y), controlling for X; path c is the total effect of X on Y; and path c′ is the direct effect of X on Y, controlling for the mediator. The indirect effect (a × b) quantifies the mediation effect and is reported as a coefficient and as a percentage of the total effect in parentheses. Model 3 included age as a covariate. Significance notation: *** *p* < 0.001, ** *p* < 0.01.

## Data Availability

The data presented in this study are available upon request from the corresponding author.

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
