# Peer review of "Sleep Quality and Patient Activation in Chronic Disease: A Cross-Sectional Mediation Analysis"

_2624-5175, 2025, doi:10.3390/clockssleep7030044_

Round 1
Reviewer 1 Report
Comments and Suggestions for Authors
1. What is the main research question addressed?
The main objective of this study is to examine the association between patient activation and sleep quality, variations between chronic disease groups, and whether chronic disease mediates this relationship..
2. Which parts do you consider original or relevant to the field?
It should be noted that subjective sleep quality, as measured by the PSQI, does not always linearly reflect the clinical burden of disease.
What specific gap in the field does the article address?
It is not a longitudinal study with ethnic variance.
3. What does it add to the subject area compared to other published materials?
This article attempts to present with its mediation analyses that chronic disease partially mediated the activation relationship of sleep quality with the patient with sleep disorder, a response between 4.7% and 6.3% of the total effect.
4. What specific improvements should the authors consider regarding the methodology?
Sociodemographic characteristics of participants included age (year of birth), gender (male/female), native language (German/Italian/Ladino/more than one or others), citizenship (Italy/other country), educational level (elementary/vocational/high school/university), community and region of origin (rural/urban), and living situation (alone/with spouse, family member, parents, or children). Chronic conditions were recorded using multiple response categories, encompassing nine specific diagnoses (e.g., cardiovascular, metabolic, oncological, etc.). These variables were coded as independent dichotomous indicators, meaning that each diagnosis was treated as a separate binary variable. All health-related variables—except living alone, which was a single item—were collected in a way that allowed for multiple simultaneous indicators. I believe that assessments related to spiritual intelligence as discussed by researcher Zohar, D., & Marshall, I. (2004). Spiritual Capital: Wealth we can live by. Berrett-Koehler Publishers would help in a more accurate search.
5. Are the conclusions consistent with the evidence and arguments presented?
The conclusions of this experiment are quite interesting and consistent with the results presented. However, in many chronic conditions, particularly those in which physiological symptoms predominate, sleep disturbances appear to be unrelated to arousal levels, highlighting the need for symptom-focused approaches. Many chronic diseases are the result of overlapping conditions caused by high levels of anxiety, which can be controlled by anxiety-control exercises, which are discussed in more detail in this excellent scientific article.
6. Are the references appropriate?
I consider them reasonable. Bibliographic references in an article can always be improved.
7. Any additional comments on the tables and figures and the quality of the data.
I believe that articles should be quite transparent so that there is no confusion or future harm to patients, and that the data and results can be repeated in research with other human beings according to their genetic characteristics. I really appreciate it when tables and graphs show statistical analyses accompanied by percentage calculations, making it easier for the reader and researchers to read and understand the experiment performed.
Reviewer 2 Report
Comments and Suggestions for Authors
This study examines the relationship between patient activation and sleep quality while attempting to analyze the mediating role of chronic diseases, but the findings are not particularly encouraging. Although the data suggest a slight association between higher patient activation levels and better sleep quality, the correlation is notably weak (r = -0.12), and the mediating effect of chronic diseases accounts for only 4.7%–6.3%—a negligible impact. Furthermore, even among patients with strong self-management abilities, chronic diseases still significantly impair sleep, casting doubt on the practical effectiveness of "activation" strategies. Overall, while this study raises an issue worth attention, it does not provide strong evidence that patient activation can meaningfully improve sleep. It may still be considered for publication as a reference for peers.
Key Concerns:
- Definition Clarity: The concept of patient activation should be more clearly defined. Specifically, Patient Activation refers to a patient’s knowledge, skills, and confidence in actively engaging in their own healthcare management.
- Low Explained Variance: The adjusted R² values in the regression analyses (PAM-10 and B-PSQI scores) are notably low. Does this weak explanatory power undermine the practical significance of the findings?
- Conciseness in Discussion: The Discussion section could be more succinct, focusing on key takeaways rather than excessive elaboration.
Reviewer 3 Report
Comments and Suggestions for Authors
The authors conducted a cross-sectional study to evaluate sleep quality and patient activation in chronic disease in South Tyrol. Overall the manuscript is well-written and this is an interesting study. However, I have a few comments that I think could help strengthen the presentation of the methods and results.
- Patient activation was measured using the Patient Activation Measure (PAM-10), and sleep quality via P Brief Pittsburgh Sleep Quality Index (B-PSQI). Chronic diseases and their number were self-reported. Please describe if there are any missing data in this study. How did you deal with the missing data?
- In line 184, please indicate if the pvalue is 2-sided or one-sided.
- Please describe how the covariates were selected in the regression analyses.
- Please discuss the generalizability of your study since the study population is only from South Tyrol.
< !-- notionvc: 54398f4b-9e71-4c99-b771-6a6d2664b932 -->
Author Response
Please, see the attachment.

Round 2
Reviewer 1 Report
Comments and Suggestions for Authors
The main objective of this study is to examine the association between patient "activity" and sleep quality, variations across chronic disease groups, and whether chronic disease mediates this relationship.
I believe that initially the quality of the English in this paper must be improved to enable understanding of this article.
The experimental design, methodology, ethical approval and issues pertaining to sleep are quite interesting and are presented in a remarkably positive manner.
I believe that articles should be very transparent so that there is no confusion or harm to patients when dosimetry, data and results are extrapolated to humans. I really appreciate it when tables and graphs show statistical analyses accompanied by percentage calculations, making it easier for the reader to read and understand.
Comments on the Quality of English LanguageI believe that initially the quality of the English in this paper must be improved to enable understanding of this article.
Author Response
Reviewer Comment 1: “The main objective of this study is to examine the association between patient ‘activity’ and sleep quality, variations across chronic disease groups, and whether chronic disease mediates this relationship.”
Response:
Thank you for restating the study aim. We would like to clarify that the term "patient activation" refers to a well-established construct describing an individual’s knowledge, skills, and confidence in managing their health, as measured by the validated Patient Activation Measure (PAM-10). This is distinct from physical activity. To avoid misunderstanding, we emphasized this definition clearly in the introduction.
Reviewer Comment 2: “I believe that initially the quality of the English in this paper must be improved to enable understanding of this article.”
Response:
We appreciate the reviewer’s concern regarding the clarity of the English. The manuscript had previously been checked using the Paperpal AI language tool and was found linguistically acceptable by two independent reviewers in the first review round. Nevertheless, we have re-reviewed the full text and made additional edits to enhance clarity, conciseness, and readability. We hope these changes resolve the concern.
Reviewer Comment 3: “I really appreciate it when tables and graphs show statistical analyses accompanied by percentage calculations, making it easier for the reader to read and understand.”
Response:
Thank you for this helpful suggestion. We confirm that all relevant descriptive tables (e.g., Table 1) already include percentage values alongside absolute counts, and that regression results are presented with confidence intervals and p-values. We have reviewed the tables once more to ensure that the layout is clear and interpretable.
Reviewer 2 Report
Comments and Suggestions for Authors
Accept in present form
Author Response
We thank the reviewer for accepting the manuscript in its present form.
Reviewer 3 Report
Comments and Suggestions for Authors
The authors have sufficiently addressed my comments. Thus, I have no further comments.
Author Response

(The authors gave the same response as above.)
